# Intravesical Gentamicin: An Option for Therapy and Prophylaxis against Recurrent UTIs and Resistant Bacteria in Neurogenic Bladder Patients on Intermittent Catheterization

**DOI:** 10.3390/antibiotics11101335

**Published:** 2022-09-30

**Authors:** Elena Andretta, Raffaele Longo, Massimo Balladelli, Camilla Sgarabotto, Dino Sgarabotto

**Affiliations:** 1Urology Department, Dolo Hospital, 30031 Venice, Italy; 2Microbiology Department, Dolo Hospital, 30031 Venice, Italy; 3Anesthesia Department, Cittadella Hospital, 35013 Padova, Italy; 4Abano Infectious Diseases Outpatient Clinic, 35031 Padova, Italy

**Keywords:** Gentamicin intravesical instillation, lower urinary tract infections (UTIs), clean intermittent catheterization (CIC), resistant bacteria, off-label prescription

## Abstract

This is a retrospective study of our experience with Gentamicin intravesical instillation as therapy and prophylaxis in patients with lower urinary tract infections (UTIs) undergoing clean intermittent catheterization because of a neurogenic bladder. It is an alternative therapy when all other systemic treatments have failed as it is still an off-label prescription.

## 1. Introduction

Gentamicin intravesical instillation is an established resource for treating lower urinary tract infections (UTIs) in patients with neurogenic bladders undergoing clean intermittent catheterization (CIC). The long history of the use of antineoplastic agents as intravesical therapy against superficial carcinoma of the bladder has opened the way to intravesical antibiotic use, with the aim of achieving similar advantages over oral and parental therapy without side effects because of a low risk of systemic absorption after intravesical instillation [1]. Compared with other antibiotics such as beta-lactams, Gentamicin shows the fastest killing rates, which are conducive to intermittent catheterizations. Taking into account that antibiotics differ considerably in the rates at which they kill bacteria, the 3 h needed for Gentamicin to achieve 90% bactericidal effects fits well with intravesical instillation [2,3].

However, Gentamicin use in intravesical instillation is still an off-label therapy with undefined dose, schedule, and value as therapy or prophylaxis. Our retrospective study of 16 patients treated with intravesical Gentamicin addresses these critical points.

## 2. Methods

This work is a retrospective study based on our practices at the Urology Department of Dolo Hospital (Venice, Italy) and the Abano Infectious Diseases Outpatient Clinic (Padova, Italy). Gentamicin bladder instillations were used in 16 cases of recurrent lower UTIs between 1 January 2015 and 31 December 2021. Symptomatic UTI is defined by a patient presenting with symptoms coherent with UTI (cloudy urine, fever, chills, bladder spasms, pain, and leakage) combined with a positive urine culture with a bacterial load above 10^5^ colony-forming units/mL. Our treatment regimen was initiated for symptomatic documented lower UTIs either when they were too frequent (defined as ≥2 UTI episodes in the last 6 months or ≥3 in the last 12 months [4]) or when resistance patterns precluded an oral alternative. Hospitalization for parenteral antibiotics was avoided, and upper UTI and prostatitis were excluded. The irrigation of 80 mg of Gentamicin diluted in 20 mL of 0.9% NaCl was performed before catheter removal during the last CIC of the day. The instillation was given twice a day for 4–7 days in the treatment regimen followed by once a day for a month and, then, every other day for 6 months in the prophylactic regimen. Out-patient follow-up every 8 weeks was continued for 6 months after completion of our protocol. The study variables included age, gender, pattern of antibiotic resistance, and the index of pathogen frequency. As a control group, we used 7 patients treated with 80 mg of intramuscular Gentamicin twice a day for 6 days. They were chosen among all patients and later also started intravesical Gentamicin. Creatinine clearance was above 30 mL/min for the patients treated with Gentamicin instillation and above 45 mL/min for the control group. Descriptive statistics methods were applied to determine the distribution of frequencies using Microsoft Excel 2019.

## 3. Results

A total of 16 patients with recurrent lower UTIs were considered. Table 1 shows the patient and control group features. The age distribution was similar between the two groups, while there was a male prevalence in gender, perhaps related to the frequency of spinal cord injury. The control group consisted of seven patients with UTIs, with four having *Pseudomonas aeruginosa Ciprofloxacin resistance* and three having *ESBL Klebsiella pneumoniae*, who were treated with intramuscular Gentamicin for 6 days. Subsequently, these latter patients had other UTIs and were started on intravesical Gentamicin. Intramuscular Gentamicin was temporarily successful in all cases, but unfortunately, all patients had recurrent UTIs within 64 days with either the same or a different microorganism.

Table 2 illustrates the frequency of recurrent UTIs, the use of antibiotics, and the type of microorganisms isolated in our patients before starting Gentamicin instillation. Our 16 patients had 64 UTIs in the 12 months before starting Gentamicin instillations, and they received 3.5 (range 2–12) courses of antibiotics for a total of 28 (range 2–74) treatment days and an average duration of 7–10 days. We found no differences between males and females in the use of antibiotics. Oral antibiotics had formerly failed several times in 46.8% of cases because of resistant bacteria (mostly *Extended-Spectrum Beta-Lactamase (ESBL) Enterobacteria and Pseudomonas Multidrug Resistant (MDR) bacteria*). Resistance to Ciprofloxacin (64.2%) and Cotrimoxazole (67.1%) was widespread, while Gentamicin resistance was marginal (3.1%).

Most patients were treated according to the above protocol; however, some patients (2; 12.5%) continued for longer periods because they were unwilling to stop their successful instillations. The median duration of Gentamicin instillation was 28 weeks (range 20–72 weeks). Fourteen (87.5%) patients were UTI-free during their Gentamicin intravesical instillation while the remaining two had a 50% reduction in their UTI frequencies during the 6-month prophylaxis and stopped intravesical Gentamicin due to clinical inefficacies. The four breakthrough infections were treated with oral (50%) or intravenous (50%) antibiotics. Strikingly, nine (56.2%) patients resumed intravesical Gentamicin off label as a continued prophylaxis or as a restart treatment after one or two UTIs.

A comparison between the patients treated with Gentamicin instillation and the control group with intramuscular Gentamicin gave similar results initially; however, within 2 months, all seven patients of the control group had recurrent UTIs and were switched to Gentamicin instillations.

Gentamicin resistance during or after the protocol was only seen in one patient. Intravesical Gentamicin, being an off-label use, cost each patient approximately EUR 285 for the entire prescription (on average, 110 ampoules) at most pharmacies in Italy.

## 4. Discussion

Clean intermittent transurethral catheterization (CIC) is an effective treatment in patients with lower urinary tract dysfunction due to neurogenic bladders, with few complications and excellent long-term results. The aim of completely emptying the bladder at regular intervals prevents vesicoureteral reflux and decreases the risk of urinary tract infections (UTIs). Nevertheless, some patients experienced frequent symptomatic UTIs with cystitis diagnosed in 29.2–36.4% of neurogenic bladder patients annually [5] and needed repeated oral antibiotic therapies. Unfortunately, drug allergies and bacterial resistance limited this approach.

Gentamicin intravesical instillation—after other methods have failed—is an established treatment in aminoglycoside-sensitive bacterial UTI. Moreover, it is now well-known that no appreciable amount of Gentamicin enters systemic circulation by this route [1]. Doses of 80 mg of Gentamicin per instillation have been found to be effective at reducing the frequency of UTIs in most studies [6,7,8,9]. Our approach was to start with a loading schedule of 80 mg of Gentamicin every 12 h for a few days in symptomatic patients. We used 80 mg of Gentamicin diluted in 20 mL of 0.9% NaCl. A rather variable volume of 80 mg of the Gentamicin solution has been reported in the literature ranging from 10 to 60 mL [6,10]. In addition, the total number of instillations during the treatment phase varies from 6 to 73. Chernyak and Salamon [8] treated 12 patients with instillations twice a week for 3 weeks, whereas Stalenhoef et al. [9] prescribed instillations to 63 subjects once a day for 2 weeks, followed by instillations every other day for 10 weeks and then twice weekly for 12 weeks. From all of these data, it is clear that the optimal number of Gentamicin instillations in this setting has not been defined. According to published reviews [6,10], at least six instillations of intravesical Gentamicin per patient should be performed to reach significant effectiveness. Of importance, the occurrence of resistance to Gentamicin despite several months of bladder instillation exposure is low, and perhaps the rapid killing rates of Gentamicin [4] and the high urinary concentration reached by Gentamicin in the bladder [6] may account for it.

We found neither Multidrug-Resistant (MDR) organisms such as *Klebsiella pneumoniae Carbapenamase-resistant* (*KPC*) bacteria among the bacterial isolates nor *Acinetobacter* or *Pseudomonas aeruginosa MDR*; thus, there was no need to consider Colistin intravesical instillation, as reported by some authors [11,12].

Comparing intravesical and intramuscular Gentamicin, as performed with our control group, highlights that only intravesical Gentamicin makes a long-term effective treatment without nephrotoxicity possible due to the absence of bladder absorption [1].

Intravesical Gentamicin has been shown to successfully treat recurrent UTIs—mostly in female—outside settings with neurogenic bladders [10]. However, this population is highly different as bladder catheterization was performed only for Gentamicin administration, while patients with neurogenic bladders underwent CIC 4–5 times a day throughout their life, thus representing a much more demanding setting with regard to UTI prevention. Therefore, intravesical Gentamicin use is not reviewed altogether for cases with [6] and without [10] neurogenic bladders.

Despite the use of Gentamicin intravesical instillation dates back to the late eighties, it is still an off-label therapy, owing to the sporadic use of these treatments. In fact, available observational studies—as reported by Pietropaolo [6]—have included a small number of patients (187 total patients in 30 years in seven studies, with 26 patients in each study on average) in the range of our own case series.

The use of Gentamicin for intravesical instillation could be made more available and less costly through hospital pharmacies, which could cut cost down to about EUR 60 [13], almost five times less than the EUR 285 each patient pay at private pharmacies. This is supported by the experience of the Medicare fee service program providing an average cost of the whole cycle intravesical Gentamicin of USD 82.90 [10].

### Study Limitations

(a)Small number of patients;(b)Discrepancy in the therapy duration between the intravesical Gentamicin group and the control group.

## 5. Conclusions

Intravesical Gentamicin instillation seems to be a safe and effective method for the treatment and prophylaxis of recurrent UTIs in patients performing clean intermittent catheterization. It is an alternative therapy when all other systemic treatments have failed even if it is still an off-label therapy.

## Figures and Tables

**Table 1 antibiotics-11-01335-t001:** Patient and control group characteristics.

	Patients	Control Group
Age (median)	57 years (range 31–78)	50 (35–62)
Gender		
Male	10 (62.5%)	5 (71.4%)
Female	6 (37.5%)	2 (28.6%)
Neurological condition		
Spinal cord injury		
Thoracic	10 (62.5%)	4 (57.2%)
Lumbar	2 (12.5%)	
Multiple sclerosis	4 (25%)	3 (42.8%)

**Table 2 antibiotics-11-01335-t002:** Use of antibiotics and UTI characteristics before initiation of Gentamicin instillations.

Symptomatic UTI median (range)	4 (2–12)
Courses of antibiotics	3.5 (2–12)
Days of antibiotic therapy	28 (5–74)
**Multidrug-resistant organisms** *ESBL Enterobacteria and Pseudomonas MDR*	30/64 (46.8%)
Ciprofloxacin resistance	41/64 (64.2%)
Cotrimoxazole resistance	43/64 (67.1%)
Gentamicin resistant organisms	2/64 (3.1%)
**Organisms on all cultures**	
*Pseudomonas aeruginosa MDR*	5/64 (7.8%)
*Klebsiella pneumonia*	8/64 (12.5%)
*Klebsiella pneumonia ESBL*	10/64 (15.6%)
*E. coli*	13/64 (20.3%)
*E. coli ESBL*	15/64 (23.5%)
*Enterococcus faecalis*	4/64 (6.3%)
*Enterobacter cloacae*	2/64 (3.1%)
*Proteus mirabilis*	2/64 (3.1%)
Other organisms	3/64 (4.7%)
Multiple organisms	2/64 (3.1%)

## Data Availability

The data presented in this study are available from corresponding author upon the request. The data are not publicly available due to privacy reasons.

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
