# Peer review of "Intravesical Gentamicin: An Option for Therapy and Prophylaxis against Recurrent UTIs and Resistant Bacteria in Neurogenic Bladder Patients on Intermittent Catheterization"

_antibiotics, 2022, doi:10.3390/antibiotics11101335_

Round 1

Reviewer 1 Report

Congratulations for your work. The study is of great interest these days.

Author Response

Thanks a lot for your appreciation and support

Reviewer 2 Report

This retrospective study conducted in 16 cases of lower UTIs treated with intravesical gentamicin between 1st January 2015 and 31st December 2021.

The topic is important and interesting. However, in my opinion the paper has some shortcomings in regards to some analyses and text, and I feel the data have not been utilized to their full extent.

Major comments:

How was the recurrent UTI defined?

Bacterial names should be italicized throughout the text.

Numbers and percentage should be reported as calculation respect to the total numbers throughout the manuscript.

Table 2 should be reported in detail. It is not clear how reported numbers  correspond to the symptomatic UTI median (range), courses of oral antibiotics, and days of antibiotic therapy.

Is there any significant differences between male and female in using antibiotics and UTI characteristics before initiation of Gentamicin instillations?

Courses of oral antibiotics, days of antibiotic therapy should be reported as mean ± SD number of UTIs in preceding of months and years.

As far as I understood, multidrug-resistant organisms (ESBL Enterobacteria and Pseudomonas MDR) were identified in 30/64(46.8%), but the reported number of individual ESBL producing bacteria is different.

The small number of patients is a significant limitation of the study that make it impossible to analyze their scale and significance values.

As stated by the authors discrepancy in the therapy duration between intravesical gentamicin group and the control group.

Actually, number of UTIs in 6 months before, during and after gentamicin instillations should be monitored and reported.

Reviewer 3 Report

The brief report submitted by Andretta E. and colleagues, it’s a retrospective study of their experience with Gentamicin intravesical instillation as therapy and prophylaxis in patients with lower urinary tract infections (UTIs) undergoing clean intermittent catheterization because of neurogenic bladder.

I have the following questions suggestions to the authors:

Table 1. It should be cited in methods, because it describes the characteristics of the patients and the controls, in addition, in the control group column, only the number of patients is observed, but not the percentages as in the patient column. In the column of patients with neurological conditions, the sum of these patients is 28 and it describes that there are only 16.

Table 2. I suggest adding a column so that the results shown are easier to understand, in one show the median (range), and in the other the frequency (percentage).

L74 mentions “The median duration of Gentamicin instillation was 28 weeks (range 20-72), however, in table 2 it is mentioned that the median is 28 days.

References. The reference data must be homogenized because some authors are underlined.

Round 2

Reviewer 2 Report

As I already mentioned, manuscript is submitted as a brief report and there are some limitations both in reporting the data and sample size that make the study insignificant. Therefore, I can not find any advantageous messages from this report even after revision.